# Reassessing the genetic variability of *Tectona grandis* through high-throughput genotyping: Insights on its narrow genetic base

Isabela Vera dos Anjos[1], Thiago Alexandre Santana Gilio[2]*, Ana Flávia S. Amorim[1], Jeferson Gonçalves de Jesus[1], Antonio Marcos Chimello[3], Fausto H. Takizawa[4], Kelly Lana Araujo[3], Leonarda Grillo Neves[3]

1 Universidade Federal de Mato Grosso, Cuiabá, Mato Grosso, Brazil, 2 Universidade Federal de Mato Grosso, Institute of Agriculture and Environmental Sciences, Sinop, Mato Grosso, Brazil, 3 Plant Genetic Improvement Laboratory, Universidade do Estado de Mato Grosso, Cáceres, Mato Grosso, Brazil, 4 Teak Resources Company, Cáceres, Mato Grosso, Brazil

☉ These authors contributed equally to this work.
* thiago.gilio@ufmt.br

**Data Availability Statement:** All relevant data for this study are publicly available from the Zenodo

## Abstract

Teak (*Tectona grandis* Linn. f.) is considered one of the most expensive hardwoods in the world. The dispersion of the species over the years has taken the teak beyond its first sources of diversity and little is known about the genetic origin and genetic variability. Thus, this study aimed to investigate the genetic diversity and genetic population structure existing in a representative teak germplasm bank collection. DNA was extracted from young leaves and each sample were genotyped by whole genome sequencing at 3 giga bases per sample, the sequences are aligned using the genome, and SNPcalls and quality control were made. To study the population structure of the genotypes, Bayesian variational inference was used via fastStructure, the phylogenetic tree was based on the modified Euclidean distance and the clustering by the UPGMA hierarchical method. Genetic diversity was analyzed based on the pairwise genetic divergence (*Fst*) of Weir and Cockerham. Genotyping by sequencing resulted in a database of approximately 1.4 million of variations SNPs were used for analysis. It was possible to identify four populations with considerable genetic variability between and within them. While the genetic variability in teak is generally known to be narrow, this study confirmed the presence of genetic variability scale in teak, which is contrary to what was initially expected.

## Introduction

Teak (*Tectona grandis* Linn. F) belongs to the family Lamiaceae, and the genus *Tectona* [1] has incomparable wood quality, which makes it a valuable wood with high demand in luxury markets, whether for construction, or furniture [2, 3]. It has also become popular due to characteristics such as the durability and workability of the wood [4].

repository (https://doi.org/10.5281/zenodo.8171369).

**Funding:** This work was funded by Fundação de Amparo à Pesquisa do Estado de Mato Grosso (FAPEMAT.0323473/2021), awarded to TG, Coordenação de Aperfeiçoamento de Pessoal de Nível Superior (88882.429503/2019-1), awarded to IV, and Teak Resources Company (TRC Agroflorestal) LTDA, awarded to LN. The funders had no role in study design, data collection and analysis, decision to publish, or preparation of the manuscript.

**Competing interests:** The authors have declared that no competing interests exist.

Considered one of the most expensive hardwoods in the world, teak logs can cost from 300 to 1000 US dollars per cubic meter depending on the quality of the log [5]. With the growing market demand, there has also been a decrease in the availability of natural plantations of teak, boosting investment in commercial plantations. It is estimated that more than 6 million hectares of teak are planted worldwide [3, 6].

Teak is a naturally occurring species in India, Myanmar, Thailand and Laos [2, 7, 8]. It is estimated that the species was introduced and naturalized in Indonesia at the beginning of the 14th century. It is believed that the teak introduced to the region of Java [2, 9–11] was imported from southern India during the period of Hinedunization [9] and, it was probably occurred because teak was initially planted with religious significance around temples [11–13]. Furthermore, there have been reports indicating that teak was brought to Indonesia from Laos and eastern Thailand [11].

In the West African region, teak was initially planted on a large scale in the early 1900s in the Ghana region [14, 15]; in 1926, it was planted in Côte d'Ivoire in national parks using seeds from Togo [16]; and in 1932, it was planted in Casamance, Senegal [11].

Teak was initially introduced in Central America, the Caribbean region, Venezuela, and Colombia in the nineteenth century, apparently between 1913 and 1916, in the region of Trinidad and Tobago with material from Myanmar (Burma at the time), in Panama in 1926 with material from Sri Lanka and in Honduras in 1927 with material from Trinidad and Tobago [17].

Over time, teak was disseminated throughout tropical Asia, tropical Africa, some Latin American countries, and the Caribbean and can also be found in some islands of the Pacific Ocean and northern Australia [2]. Currently, approximately 70 countries throughout tropical Asia, Africa, Latin America, and Oceania have teak plantations [3, 6]

In Brazil, teak planting began in the late 1960s in the region of the city of Cáceres in the state of Mato Grosso by the company Cáceres Florestal SA [18]. Since then, it has been among the main forest species planted in Brazil. According to the last report released by the Brazilian Tree Industry (IBÁ) in 2019, the planted area of teak increased from 65 thousand ha (2010) to approximately 93 thousand ha [19].

The exact origin of teak is still unclear, as there are almost no records of its origin, and many plantations were established using local trees. This lack of knowledge generates a challenge for researchers [11]. Over time, the genetic diversity of teak has been investigated in several studies with different techniques, such as using isoenzymes as molecular markers [20] and random amplified polymorphic *DNA* (RAPD) [21, 22], amplified fragment length polymorphisms (AFLPs) [23–26], microsatellite markers [27–29], and single nucleotide polymorphisms (SNPs) [13, 30, 31].

Although the teak reference genome is not yet available, drafts of this genome are available. The first estimate, created by flow cytometry, estimated a genome with 18 pseudochromosomes with 465 Mbp [32]; subsequently, using complete genome sequencing technology based on next-generation sequencing (NGS-WGS), the genome was estimated at 317.5 Mbp [33]. Most recently, 19 pseudochromosomes and a genome size of 338 Mbp have been estimated via long-read sequencing [34]. These previous studies were responsible for several advances in the genetic improvement of teak, the use of available genetic resources and advances in biotechnological development and prospecting in teak.

The germplasm conservation and collection is important toll to deal with challenges as genetic vulnerability, diseases resistance, environmental/climate stress, wood quality and production. Also, a good poll of genetic resources is the base for genetic breeding programs and the development of new cultivars which can change the economic at one or more regions. Thus, the current study aims to use high-throughput genotyping to investigate the genetic diversity, genetic variability, and the population genetic structure, in a germplasm bank

located in Brazil. Furthermore, these study tries to suggest groups of diversity as well as the relationship between and among populations.

## Materials and methods

### Teak germplasm bank

The studied population was obtained from an in vivo germplasm bank from Teak Resources Company accessed by the Universidade do Estado de Mato Grosso (UNEMAT)—Campus Cáceres city. A total of 241 different genotypes of *Tectona grandis* Lf from an in vivo germplasm collection situated in Rosário d'Oeste city in Mato Grosso state, Brazil, were used in this study. This collection comprises genotypes from different origins, including Honduras, India, Indonesia, Thailand, Malaysia, and Myanmar, as well as genotypes collected from seminal teak fields in 1995 in Brazil. Some genotypes' origins are registered as mixed (India/Thailand/Laos/Indonesia/Africa—collected in a 1995 origin test [35, 36]), while those of some unregistered genotypes are treated as unknown. The origin of each genotype is available in S1 Table.

### DNA extraction, preparation, and genotyping by genome wide sequencing

To obtain DNA samples of 241 genotypes, young leaves of only one plant per genotype in vivo germplasm bank were collected, identified and immediately stored in liquid nitrogen for subsequent maceration. DNA extraction for each sample were performed according to the CTAB extraction protocol [37] with few modifications (CTAB at 4% and 2-mercaptoethanol at 3%) the samples were purified and concentrated with the Genomic DNA Clean & Concentrator Kit (Zymo Research). Subsequently, these samples were measured with the QuantiFluor One dsDNA System Kit (Promega) in Qubit 2.0 and standardized to ensure that all the samples contained 10 ng/μl of DNA. These samples were then lyophilized before being sent for genotyping.

The samples were genotyped by means of complete genome sequencing (whole genome sequencing, WGS) with 8X coverage (3 giga bases per sample. This step was performed by the company SNPsaurus, Oregon, USA. Samples were prepared using the DNA Flex kit (Illumina Inc, San Diego) with 5 ng of genomic DNA used for input. Sequencing libraries were run on multiple S4 lanes of a Novaseq 6000 with paired-end 150 bp reads (GC3F, University of Oregon). Demultiplexed sequence reads were aligned to the teak_tectona_grandis_26-Jun2018_7GlFM_fmt_tp.fa reference (https://zenodo.org/record/3962665#.Y7MxfezMKrM) using bbmap (BBTools) with the parameters minid = .95 and ambig = toss. The resulting bam files were converted into a vcf format genotype table using callvariants (BBTools) with the parameter's ploidy = 2 multisample = t nopassdot = f minad = 2 minavgmapq = 15 minreadmapq = 15.

The data obtained from genotyping then underwent quality control (QC) to ensure greater reliability and were submitted to filtering using the PLINK v1.90b6.24 package [38] with the R v.4.1.3 software [39] and RStudio [40]. Filtering was performed for minimum allelic frequency (MAF>0.1) using the "maf" function, for call rate (>0.1) using the "mind" function, and for quality of genotyping (QG>0.1) using the "geno" function. In view of the volume of SNPs obtained, in this study we chose to be rigorous in terms of filtering, dispensing with any imputation process. Based on work already carried out that reported greater efficiency of markers using these indices as a cut [41, 42]. Also, the "rMVP" package [43] was used to create the distribution graph of SNPs by chromosome.

## Population structure, phylogenetic tree and principal component analysis

To identify the population groups among the genotypes, we used fastStructure v1.0 software [44], which uses Bayesian variational inference. fastStructure was performed in the standard configuration, and the convergence criterion used was 10x10-6, with K ranging from K = 1 to 10. To choose the number of clusters, the function "chooseK.py" was included in the fastStructure package for K = 1–10, and the results of the best K were represented graphically using the "distruct.py" function also included in the fastStructure package.

The UPGMA method [45] was used to infer evolutionary history, the bootstrap consensus tree was inferred from 500 replicates [46]. Branches corresponding to partitions reproduced in less than 50% of bootstrap replicas are collapsed. The evolutionary distances were calculated using the Maximum Composite Likelihood method [47] and are in units of the number of base substitutions per site. All ambiguous positions were removed for each pair of sequences (paired deletion option). Evolutionary analyzes were performed on MEGA11 [48].

Principal component analysis (PCA) was performed using the PLINK v1.90b6.24 package [38] with the R v.4.1.3 software [39] and RStudio [40] to calculate the genetic distances between individuals based on identity by state (IBS), i.e., if the individuals had identical nucleotide sequences. Subsequently, PCA was performed using the "*cmdscale*, *cbind*, and *round*" functions of RStudio. To visualize the PCA results, the packages Tidyverse v.1.3.2 [49] and ggplot2 v.3.36 [50] were used.

## Genetic diversity of *Tectona grandis* Lf genotypes

For the genetic diversity analyses, the clusters defined by the population structure analysis with the clusters resulting from the fastStructure analysis were used. The *vcftools* v. 0.1.16 program [51] was used to calculate the pairwise genetic divergence (*Fst*) among the clusters according to methods described by Weir and Cockerham [52]. For the interpretation of the *Fst* indices, the following classification was used: *Fst* values <0.05 indicate little differentiation between populations, values between 0.05 and 0.15 indicate moderate differentiation, values between 0.15 and 0.25 indicate strong differentiation and values > 0.25 indicate very strong differentiation [53, 54].

The observed and expected total heterozygosity rates (HoT and HeT, respectively) and the observed and expected heterozygosity per population (HoP and HeP, respectively) were obtained with *vcftools*, and the proportion of missing data for all teak genotypes was obtained using the "*Geno Summary*" function of TASSEL software [55].

# Results

## Data sequencing

We used WGS to genotype by sequencing the 241 teak genotypes, this resulted in over 5 million SNPs. After the data filtering process, 1,446,216 SNPs and 211 genotypes with call rate >90% were selected; MAF <90%, QG <90% for use in the study, thus ensuring that these markers were consistent and had high reproducibility, the distribution of these SNPs by pseudo chromosomes can be seen in Fig 1.

The markers showed diversity, with a mean allele frequency of the reference allele of 0.82 and a mean allele frequency for the allele with the alternative allele of 0.17. The mutations with the highest frequencies observed were G to A and C to T, referring to substirutions and transitions , both with 31%, and the mutation with the lowest observed frequency was transversion (C/G), with 7%, as shown in Fig 2. We also observed two transversion mutations with 9% (T/G

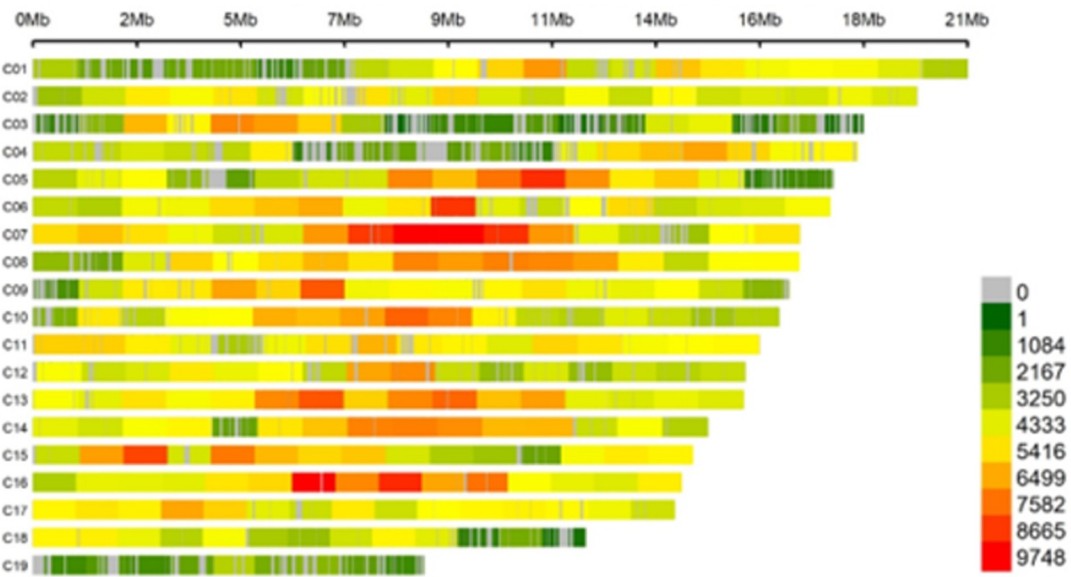

**Fig 1. Density and distribution of SNPs variants across the teak genome in 211 *Tectona grandis* Lf genotypes.**

and A/C) and one with 13% (T/A). In general, the 1,446,216 high-quality SNPs were distributed among the 19 pseudo chromosomes of teak.

## Population structure results

The genetic population structure analysis by Bayesian inference in fastStructure software determined two possible clustering for the 211 genotypes, with first peak of significance at four K cluster and second peak at five K clusters, as shown in Fig 3. The respective origins and number

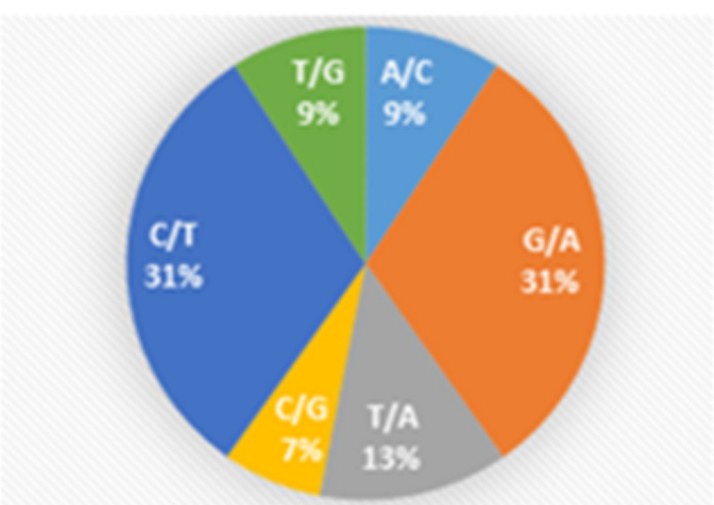

**Fig 2. Types of SNP variants observed in the genotypes of *Tectona grandis* Lf.**

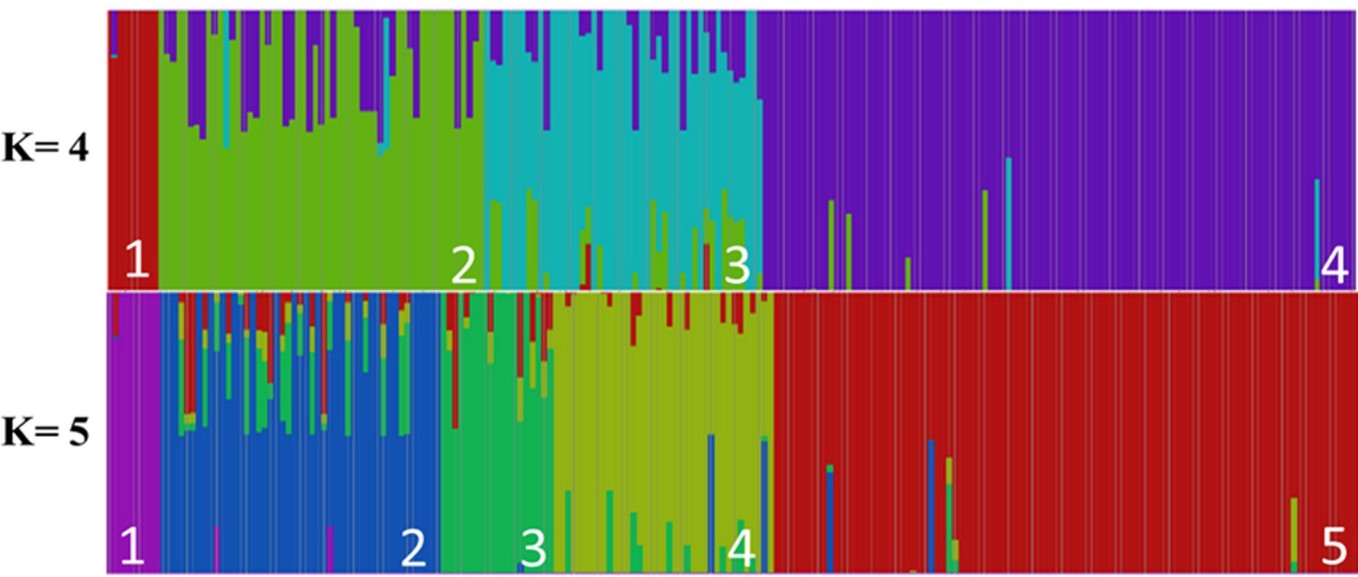

**Fig 3. Genetic population structure of 211 distinct genotypes of *Tectona grandis* Lf.** Representing the two potential clustering patterns produced by the analysis carried out using the fastStructure software (K = 4 and 5). The numbers in white color is indicating each grouping structure.


of individuals in each group are described in Table 1, for better visualization, as already mentioned, the list of genotypes as well as their respective origins are available in S1 Table.

According to the genetic population structure results and based on the origins of the genotypes, the K = 4 cluster better fitted the data; thus, the 211 teak genotypes presented four possible clustering, thus suggesting four subpopulations. As shown in Table 1, cluster one had the lowest number of genotypes (nine genotypes), from Brazil, Honduras, Malaysia, and unknown location; genotype from Brazil showed introgression of subpopulations three and four according to can be viewed on S1 Fig and S2 Table.

Conversely, cluster two comprised 55 genotypes collected from Brazil, Thailand, Indonesia, Malaysia, and Mix. This cluster showed a higher proportion of individuals with introgressions, with a total of 30 genotypes with introgressions. Most of these introgressions occurred in cluster four, with three genotypes presenting other introgressions: genotype 0221 presented 49%

**Table 1. Number of genotypes and their respective origins in each subpopulation of the population structure graph.**

| | SPE | BRA | TAI | INDO | IND | HON | MYA | MAL | MIX | UNKN | TOTAL |
|---|---|---|---|---|---|---|---|---|---|---|---|
| K = 4 | **1** | 1 | - | - | - | 1 | - | 6 | - | 1 | **9** |
| | **2** | 9 | 22 | 12 | - | - | - | 8 | 4 | - | **55** |
| | **3** | 6 | - | - | 4 | - | - | 20 | 14 | 3 | **47** |
| | **4** | 77 | 3 | 4 | - | - | 1 | 1 | 4 | 10 | **100** |
| **K = 5** | **1** | 1 | - | - | - | 1 | - | 6 | - | 1 | 9 |
| | **2** | 6 | - | - | 4 | - | - | 20 | 14 | 3 | 47 |
| | **3** | 3 | 1 | 11 | - | - | - | 2 | 2 | - | 19 |
| | **4** | 7 | 21 | 1 | - | - | - | 6 | 2 | - | 37 |
| | **5** | 76 | 3 | 4 | - | - | 1 | 1 | 4 | 10 | 99 |

SPE–Subpopulation, BRA—Brazil, TAI-Thailand, INDO—Indonesia, IND—India, HON—Honduras, MYA—Myanmar, MAL- Malaysia, MIX—Mixture (India/Thailand/Laos/Indonesia/Africa/Mexico), UNKN- unknown origin.

introgression of cluster three, genotype 0125 presented introgressions of cluster three (4.9%) and four (46, 9%), and genotype 0135 also showed introgressions of cluster three and four in other proportions, 47.0% and 2.6%, respectively.

Cluster three comprised 47 genotypes collected in Brazil, India, Malaysia, Mix and an unknown location, and a total of 22 of these genotypes presented introgressions, mostly by cluster two and four, while two genotypes from Malaysia (0188 and 0178) and one genotype from Brazil (0009) also had introgressions of cluster one. All the genotypes of this cluster had two or more introgressions except for genotype 0005, which presented introgressions only for cluster four.

Finally, cluster four had the largest number of genotypes (100 genotypes), which were collected in Brazil, Thailand, Indonesia, Myanmar, Malaysia, Mix and an unknown location and showed few introgressions (7 genotypes). Four genotypes had only one introgression of cluster two, one genotype presented only one introgression for cluster one, and two genotypes presented introgression of cluster two and three. A total of 60 teak genotypes of the 211 studied showed some mixing proportion with other cluster.

## Results phylogenetic study

It was also possible to identify four groups by the UPGMA hierarchical clustering method: one group with 39 individuals and another large group that was divided into three subgroups (with 53, 97 and 22 individuals, respectively), as shown in Fig 4. The origins and the numbers of individuals in each group are shown in Table 2.

Thus, it was possible to observe that the phylogenetic tree partially agreed with the results obtained in the population structure analysis. Most of the genotypes did not diverge in the clusters.

As it is possible to visualize, the first group of the phylogenetic tree (group one) included 39 genotypes that originated from Brazil, India, Honduras, Malaysia, Mix and unknown locations, with the largest portion of the genotypes of this group represented by genotypes from Malaysia (18 genotypes). Cluster one of the phylogenetic tree corresponds to cluster one of the population structure, with small differences between groups.

The second grouping (group two) included 53 genotypes originating from Brazil, Thailand, Indonesia, Malaysia and Mix. Most of the genotypes included in this subpopulation were from Thailand (22 genotypes), and this grouping corresponded to cluster two, diverging in one genotype that belonged to cluster four (0024) and excluding two genotypes that were part of cluster two (0135, 0221).

The third group (group three) comprised the largest number of individuals, with 97 teak genotypes collected in Brazil, Thailand, Indonesia, Myanmar, Mix and unknown locations. The largest portion of these genotypes were from Brazil (76 genotypes). This group corresponded to cluster four, however, three genotypes belonging to cluster four did not remain in this cluster (0024, 0165, 0229).

The fourth group (group four) included 22 genotypes collected in Brazil, India, Malaysia, Mix and unknown locations, with the majority collected in Malaysia (ten genotypes). This group partially agreed with cluster three, with divergences already clarified above.

## Results analysis principal components

Principal component analysis (PCA) pointed that two components explained 76.06% of the variation, with PC 1 accounting for 72.59% and PC 2 accounting for 3.47% (Fig 5). The PCA resulted in a different clustering pattern with five groups; however, one of these groups included only the 0183 genotype from India.

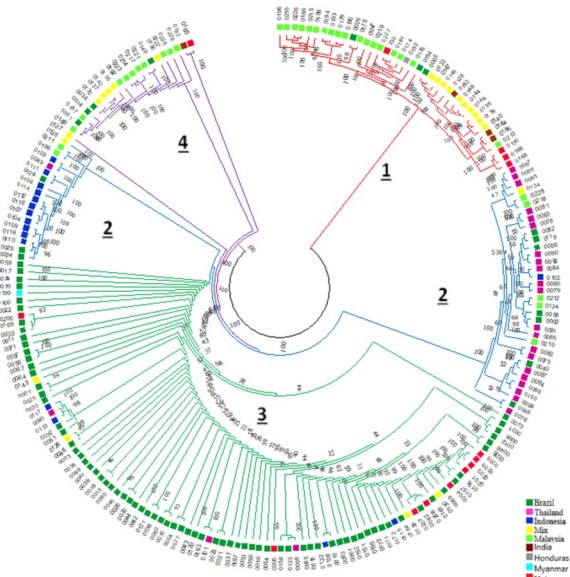

**Fig 4. Phylogenetic tree generated by the UPGMA hierarchical clustering method for the 211 genotypes of** *Tectona grandis* **Lf.** The suggested groups are indicated by the corresponding colors and numbers inside the tree.

This genotype showed introgressions, with cluster two mixing with cluster one and three at 32.48% and 5.59%, respectively. There was also a concentration of genotypes from Malaysia and some from Brazil in the two main central groups, differing from the genotypes from Indonesia contrary to the results of previous analyses. The results of the PCA analyzes are available in S3 Table.

## Genetic divergence results

Greater reliability in the choice of clusters was obtained through the analysis of genetic divergence by the mean *Fst* estimate described by Weir and Cockerham [52]; estimates were performed between of the clusters produced by fastStructure (K = 4 and 5), the *Fst* values were produced between all the clusters. The cluster are described in Table 3.

These indices confirm that the grouping with four cluster (K = 4) was the one that best fit the data with the highest *Fst* values, indicating moderate and very strong differentiation between the subpopulations. The lowest *Fst* values were found between cluster one and three (0.13) and between cluster two and four (0.05), representing moderate differentiation. Very strong differentiation was observed between all the other cluster.

**Table 2. Number of genotypes and their respective origins in each subpopulation according to the UPGMA hierarchical clustering method.**

| GRU | BRA | TAI | INDO | IND | HON | MYA | MAL | MIX | UNKN | TOTAL |
|-----|-----|-----|------|-----|-----|-----|-----|-----|------|-------|
| 1 | 3 | - | - | 3 | 1 | - | 18 | 10 | 4 | 39 |
| 2 | 10 | 22 | 12 | - | - | - | 7 | 2 | - | 53 |
| 3 | 76 | 3 | 4 | - | - | 1 | - | 4 | 9 | 97 |
| 4 | 4 | - | - | 1 | - | - | 10 | 6 | 1 | 22 |

GRU–phylogenetic tree groups, BRA—Brazil, TAI -Thailand, INDO—Indonesia, IND—India, HON- Honduras, MYA—Myanmar, MAL- Malaysia, MIX–Mixture (India/Thailand/Laos/Indonesia/Africa/Mexico), UNKN- unknown origin.

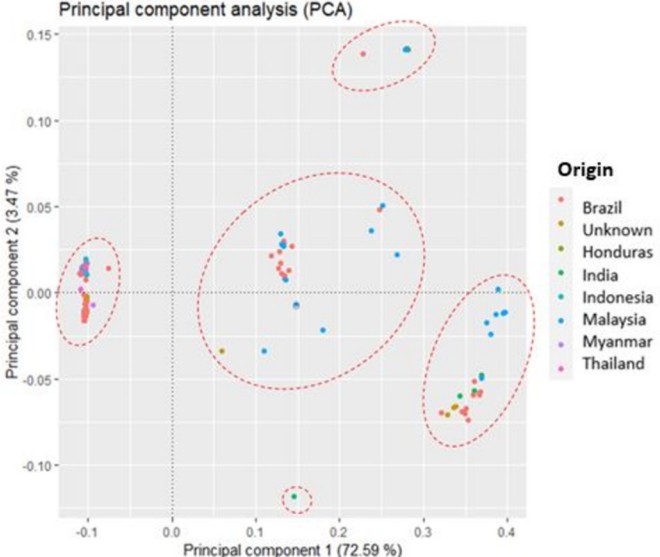

**Fig 5. Principal component analysis for the 211 genotypes of *Tectona grandis* Lf.** The colors represent their respective regions of origin.

The mean proportion of missing data was 0.01, and the mean total observed heterozygosity (*HoT*) was 0.18, whereas the observed heterozygosity per population ranged from 0.12 (SPE 2) to 0.40 (SPE 1) as shown in Table 4.

**Table 3. Mean estimates of Weir and Cockerham *Fst* among the subpopulations resulting from the fastStructure analysis.**

| K = 5 | | | | | |
|---|---|---|---|---|---|
| SPE | 1 | 2 | 3 | 4 | 5 |
| 1 | - | - | - | - | - |
| 2 | 0.13** | - | - | - | - |
| 3 | 0.48**** | 0.32**** | - | - | - |
| 4 | 0.46**** | 0.34**** | 0.04* | - | - |
| 5 | 0.49**** | 0.39**** | 0.05** | 0.07** | - |

| K = 4 | | | | |
|---|---|---|---|---|
| SPE | 1 | 2 | 3 | 4 |
| 1 | - | - | - | - |
| 2 | 0.46**** | - - | - - | - - |
| 3 | 0.13** | 0.36**** | - - | - - |
| 4 | 0.49**** | 0.05** | 0.39**** | - - |

SPE - Population structure subpopulations

\* - Little differentiation between subpopulations

\*\*- moderate differentiation

\*\*\* - strong differentiation

\*\*\*\* - very strong differentiation [53, 54].

**Table 4. Genetic diversity parameters of the four populations of *Tectona grandis*.**

| Subpopulation | *Ho* | *He* |
|---|---|---|
| SPE 1 | 0.4058 | 0.2699 |
| SPE 2 | 0.1232 | 0.2699 |
| SPE 3 | 0.3163 | 0.2700 |
| SPE 4 | 0.1401 | 0.2699 |

SPE - subpopulations in the population structure; *He* - expected heterozygosity; *Ho* - observed heterozygosity.

Based on the results, it possible was determined that the 211 genotypes of *Tectona grandis* genetically diverged into four distinct subpopulations (or clusters), with genetic variability between and within the four subpopulations.

## Discussion

The recent application of molecular tools such as whole-genome sequencing has created new possibilities and led to new findings, resulting in the reformulation of theories for many species, including those with less economic impact worldwide. The present study demonstrates that genotype-by-sequence true whole-genome sequencing can be used to analyze genetic diversity and population genetic structure over a significant period, with comprehensive information across the genome and with labor savings compared to other methodologies.

The germplasm bank used in this study contains genotypes from various origins, including Honduras, India, Indonesia, Thailand, Malaysia, and Myanmar, as well as genotypes collected from seminal teak fields in Brazil in 1995. This pool of genotypes represents the regions of natural occurrence of teak [7, 11, 12].

Through Bayesian variational inference analysis, the 211 teak genotypes were subdivided into four subpopulations, corroborating the hypothesis that there are four main centers of genetic variability scattered across India, Africa, Thailand, Laos, Indonesia, and Ghana. However, there is some disagreement among authors regarding the precise regions [11, 56, 57].

More recently, Wanders et al. [58] used genotyping by sequencing (GBS) to study the genetic diversity of teak in Ghana, identifying between two and four clusters. According to the authors, the most likely number of genetically distinct populations is two; however, only 37 genotypes from Thailand, and Laos were not included, and Asia was represented only by genotypes from northern India and Malaysia, which probably caused the division into only two populations.

Teak flowers are typically cross-pollinated by insects due to their hermaphrodite nature [10, 13, 36] and self-incompatibility [18, 30], making the experiment of self-pollination challenging [59]. Higher estimates of expected heterozygosity within individuals of both clusters confirm this (Table 4). This could also be explained by the fact that cross-pollinated species quickly accumulate new gene combinations and generally harbor high genetic diversity. Earlier investigations by Ansari et al. [60], Shrestha et al. [61], Vaishnav and Ansari [62] also showed that majority of variation in teak lies within a population than between populations. This has significant bearing on the conservation and management genetic resources of perennial, woody, cross pollinated forestry species such as teak.

The concern about the low genetic diversity of the species has been addressed in several studies. Although it is an outcrossing species with high rates of crossing, teak is predominantly propagated through clonal propagation, further limiting the species diversity. Artificial selection within breeding should be used judiciously, as it may contribute to genetic erosion, and it is crucial to ensure sufficient variability within a breeding program [63–65]

Low genetic diversity limits the characteristics available to be explored in a breeding program. Moreover, it can also affect the adaptability and resilience of the crop, reducing the species' chances of facing future issues and evolving over the years. Fofana [56] emphasized the need to conserve teak germplasm, considering the world's climate changes, based on the decrease of the species' natural forests and the loss of diversity caused by monoculture. Strategies aiming to conserve and increase genetic diversity are crucial to overcome these limitations and achieve success in genetic improvement programs.

Although several studies have reported a narrow genetic base for Teak, with only two [57, 58] or three [28] genetic groups, this difference can be explained using different genetic markers (SNPs and microsatellites) [66]. It should be noted that this study used the largest number of variants to date, with 1,494,216 Teak SNPs; the largest number previously used was 23,182 SNPs [58].

The genotypes collected in Brazil were present in all the proposed groups, suggesting high variability in the origin of these genotypes; unfortunately, there is not a very clear record about the introduction of teak to Brazil. However, it is known that the first introduction of teak to Brazil occurred in 1961 in the city of Cáceres, Mato Grosso, through seeds from material from Trinidad and Tobago that have records of their origin in "Tenasserim, Burma" (currently Tanintharyi, Myanmar) [66]; this strain became known as Tenasserim-Trinidad [67].

Another introduction was reported in 2003 in the city of Cuiabá, Mato Grosso, with the introduction of genetic materials called "YSG Biotech TG1-8", which are eight superior clones from breeders of the Solomon Islands developed through the breeding program in Sabah, Malaysia, by the Biotech division of *Yayasan Sabah Group* [68]. In Latin America, in general, the records suggest that initially, the introduced teak originated from Myanmar and India, and later, some genotypes originated from Thailand in 1990 [69]; however, information on this introduction is also scarce.

Based on this history and the inclusion of Brazilian genotypes in all the groups (Tables 1 and 2), it can be assumed that the origins of the genetic material currently available in Brazil are variable and that its initial introduction via seed may have generated some variability.

The largest portion of Brazilian genotypes was gathered in cluster four, group three (77 and 76 genotypes, respectively) and in cluster two, group two (9 and 10, respectively). These clusters have individuals from Thailand, Indonesia, Myanmar, and Malaysia, corroborating the available history of the species introduction to Brazil.

We can also highlight that based on these clusters, it was possible to observe that the genotypes from Thailand and Indonesia were grouped into only two groups, and with full agreement between the population structure and phylogenetic tree clusters, the largest proportions of these genotypes were included in subpopulation two, group two (22 genotypes from Thailand and 12 genotypes from Indonesia), and in the subpopulation four, group three (three genotypes from Thailand and four from Indonesia). Although these genotypes were separated into different groups, these two clusters showed the lowest genetic differentiation between subpopulations.

Although subpopulation four had the largest number of genotypes, it had a lower-than-expected heterozygosity of 0.14; the same pattern occurred for subpopulation two, with $Ho$ with 0.12, indicating that these subpopulations contained a greater number of homozygotes, leading to lower genetic variability. This most likely occurs as a result of the founding effect caused by the artificial selection of phenotypically superior trees [70], considering that teak is propagated vegetatively and usually used in monoculture, always reproducing the same genotypes of superior quality and limiting the variability in the species, as this limitation tends to decrease the variability present in the population.

Although the genetic variability within these subpopulations is considered low, when compared with the other subpopulations, there is high genetic divergence according to the *Fst* values; for example, subpopulation four showed very strong differentiation with subpopulation one (0.49) and subpopulation three (0.39) and moderate differentiation with subpopulation two (0.05), while subpopulation two also showed very strong differentiation with subpopulation one (0.46) and subpopulation three (0.36), according to Table 3.

According to the heterozygosity and genetic divergence of both groups, it was possible to conclude that cluster two and four are genetically close and have a strong genetic relationship with Thailand and Indonesia; however, the separation of these cluster occurred due to the proportion of mixed genetic origins present in several genotypes (30 individuals) of cluster two.

Considering that Thailand is a region of natural occurrence of teak, it can be assumed that the genotypes collected in Indonesia are of Thai origin. This finding supports the hypothesis that the teak in Indonesia originates in eastern and northwestern Thailand and the central region of Laos [11, 67] but not the hypothesis that the teak introduced and naturalized in Indonesia would have come from southern India through Hindunization between the 14th and 16th centuries. Seedlings were probably planted around temples since the teak was thought to have religious significance; in particular, it was believed that incarnations of the souls of ancestors resided in teak trees, and teak is still planted around temples in northern Thailand [11].

The genotypes collected in the Mix provenance test in 1995 were present in all the clusters, except for cluster one. This finding was expected because these genotypes were collected from unidentified trees; however, it is known that there were genotypes from India, Thailand, Laos, Indonesia, Africa, and Mexico present in this test [35, 36]. The largest portion of the Mix genotypes was allocated to cluster three, group four (14 and 6 genotypes, respectively); most of the genotypes from Malaysia were included in this grouping, and cluster three also included the four genotypes from India.

Unfortunately, as already mentioned, the available records on the introduction of teak outside of its natural region are scarce, as is the case with teak introduced in Malaysia. However, there are reports of the introduction of clones from the company *YSG Bioscape Sdn. Bdd.* collected from Indian origins used in the production of superior genotypes in the country [71].

This introduction may explain the strong genetic relationship found between the Indian and Malaysian genotypes in cluster three; the same relationship between the Malaysian and Indian genotypes has already been reported in another study [58]. According to the data, of the 22 genotypes in this group that showed some mixing, ten were from Malaysia. It is possible to assume that this occurred due to the propagation of superior genotypes from the company *YSG Bioscape Sdn. Bdd.* This cluster showed a higher-than-expected *Ho* of 0.31, strong differentiation with cluster two (0.36) and four (0.39), and moderate differentiation with cluster one (0.13).

Most likely, the proximity of subpopulation three to cluster one is due to the genetic relationship between trees from Malaysia and India. Given that this group included six genotypes from Malaysia, it was also possible to verify that the phylogenetic analysis was comprehensive, clustering 18 genotypes from Malaysia and three from India into group one. It can be assumed that these six genotypes from Malaysia differ from cluster three only because they do not have any mixing ratio with other populations, given that in cluster one, only one genotype from Brazil showed some introgression to 0154 and the six genotypes from Malaysia did not have any introgression and showed no mixing with other clusters.

It was also possible to infer the origins of the unknown genotypes that are present in all groups, except in cluster two, group two, based on the data showing that most of the unknown genotypes have genetic relationships with the Thailand/Indonesia genotypes.

Unfortunately, in this study, it was not possible to represent all the regions of natural occurrence of teak; for example, genotypes from Laos were not collected, only one genotype was

collected from Myanmar, and only four genotypes were collected from India. The legislation of each region should be taken into account, as some countries currently hinder the export of seeds and sometimes clones of their genetic heritage [58].

For example, in Brazil, the importation of teak seedlings from Malaysia is authorized by the Ministry of Agriculture, Livestock and Supply through the International Agricultural Surveillance System (Vigiagro) according to some criteria; however, review of the ARP system (Vegetbale Products of Authorized Import) resulted in importation from Laos, Myanmar and Thailand being blocked [72].

There is genetic diversity within and between populations, with particular emphasis on clusters one and three, within which greater variability was observed (*Ho* of 0.41 and 0.32, respectively). However, the variability between these clusters is considered moderate, with an *Fst* of 0.13. The diversity within these clusters supports the hypothesis that India can be considered a center of teak diversity [28].

The greater differentiation between cluster one and four (0.49) can be explained by the strong relationship between the Indian-Malaysian genotypes in cluster one and the Thai-Indonesian genotypes in cluster four. It is known that genetic variability is extremely important for conservation or breeding programs of the species, providing the possibility of genetic mixtures to maximize possible genetic gains [11].

## Conclusions

For many years, studies have suggested that the genetic variability in *Tectona grandis* was restricted and with a possible Bottle neck structure; however, our data indicates that the genetic variability is larger than what most studies have reported. This study allowed for the detection and characterization of genetic diversity in a field germplasm, highlighting the need for integration of a representative pool of genotypes from a germplasm bank. The data from this study will be useful for genetic improvement and conservation programs at both regional and global levels.

## Supporting information

**S1 Fig. Population structure of the 211 *Tectona grandis* genotypes for K = 4.** The numbers on the left of the image correspond to the identification of each genotype.
(TIF)

**S1 Table. Registered origin of the 211 *Tectona grandis* Lf genotypes used in this study.**
(XLSX)

**S2 Table. Introgression data from the population genetic structure analysis of the 211 *Tectona grandis* Lf genotypes for K = 4.**
(XLSX)

**S3 Table. Data from the five main components analyzed in the 211 genotypes of *Tectona grandis* Lf.**
(XLSX)

**S1 File. This is the scripts used in Rstructure.**
(R)

## Acknowledgments

We thank the Teak Resources Company (TRC), as well as the State University of Mato Grosso (UNEMAT) for their support and assistance with material and infrastructure, as well as the Federal University of Mato Grosso (UFMT) for providing us with training opportunities. The second author extends gratitude to the Institute of Agricultural and Environmental Sciences (ICAA), UFMT, Sinop, MT, Brazil.

## Author Contributions

**Conceptualization:** Isabela Vera dos Anjos, Thiago Alexandre Santana Gilio, Antonio Marcos Chimello, Fausto H. Takizawa, Kelly Lana Araujo, Leonarda Grillo Neves.

**Data curation:** Isabela Vera dos Anjos, Thiago Alexandre Santana Gilio, Ana Flávia S. Amorim, Jeferson Gonçalves de Jesus, Antonio Marcos Chimello, Fausto H. Takizawa, Kelly Lana Araujo, Leonarda Grillo Neves.

**Formal analysis:** Isabela Vera dos Anjos, Thiago Alexandre Santana Gilio.

**Funding acquisition:** Fausto H. Takizawa, Kelly Lana Araujo, Leonarda Grillo Neves.

**Investigation:** Isabela Vera dos Anjos, Thiago Alexandre Santana Gilio, Antonio Marcos Chimello, Kelly Lana Araujo, Leonarda Grillo Neves.

**Methodology:** Isabela Vera dos Anjos, Thiago Alexandre Santana Gilio.

**Project administration:** Thiago Alexandre Santana Gilio, Kelly Lana Araujo, Leonarda Grillo Neves.

**Resources:** Isabela Vera dos Anjos, Thiago Alexandre Santana Gilio, Leonarda Grillo Neves.

**Software:** Isabela Vera dos Anjos, Thiago Alexandre Santana Gilio.

**Supervision:** Thiago Alexandre Santana Gilio, Leonarda Grillo Neves.

**Validation:** Isabela Vera dos Anjos, Thiago Alexandre Santana Gilio.

**Visualization:** Isabela Vera dos Anjos, Ana Flávia S. Amorim, Jeferson Gonçalves de Jesus.

**Writing – original draft:** Isabela Vera dos Anjos.

**Writing – review & editing:** Thiago Alexandre Santana Gilio.

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
