## [Decision Letter · Decision Letter 0]

18 Jul 2023

PONE-D-23-12305Reassessing the Genetic Variability of Tectona grandis through high-throughput genotyping: Insights on its Narrow Genetic BasePLOS ONE

Dear Dr. Gilio,

Thank you for submitting your manuscript to PLOS ONE. After careful consideration, we feel that it has merit but does not fully meet PLOS ONE’s publication criteria as it currently stands. Therefore, we invite you to submit a revised version of the manuscript that addresses the points raised during the review process.

We look forward to receiving your revised manuscript.

Kind regards,

Mehdi Rahimi, Ph.D.

Academic Editor

PLOS ONE

Journal Requirements:

"This research was financially supported by the Mato Grosso Research Support Foundation (FAPEMAT) and the Coordination for the Improvement of Higher Education Personnel (CAPES) through grants. We also extend our gratitude to the Teak Resources Company (TRC) and the University of the State of Mato Grosso (UNEMAT) for their assistance with material and infrastructure, as well as the Federal University of Mato Grosso (UFMT) for providing us with training opportunities. The second author extend the gratitude to the Institute of Agriculture and Environmental Sciences (ICAA), UFMT, Sinop, MT, Brazil."

"This study was financed in part by the Coordenação de Aperfeiçoamento de Pessoal de Nível Superior – Brasil (CAPES) and by Teak Resourses Company (TRC agroflorestal) LTDA"

5. We note that you have stated that you will provide repository information for your data at acceptance. Should your manuscript be accepted for publication, we will hold it until you provide the relevant accession numbers or DOIs necessary to access your data. If you wish to make changes to your Data Availability statement, please describe these changes in your cover letter and we will update your Data Availability statement to reflect the information you provide."

6. We note that you have referenced (Keogh R. Teak (Tectona grandis) provenances of the Caribbean, Central America, Venezuela and Colombia. Unpublished Original document at FAO headquarters, Rome. 1978.) which has currently not yet been accepted for publication. Please remove this from your References and amend this to state in the body of your manuscript: (ie “Bewick et al. [Unpublished]”) as detailed online in our guide for authors

7. We note that Figure 1 in your submission contain map images which may be copyrighted. All PLOS content is published under the Creative Commons Attribution License (CC BY 4.0), which means that the manuscript, images, and Supporting Information files will be freely available online, and any third party is permitted to access, download, copy, distribute, and use these materials in any way, even commercially, with proper attribution. For these reasons, we cannot publish previously copyrighted maps or satellite images created using proprietary data, such as Google software (Google Maps, Street View, and Earth). For more information, see our copyright guidelines: http://journals.plos.org/plosone/s/licenses-and-copyright.

Reviewers' comments:

Reviewer's Responses to Questions

**Comments to the Author**

1. Is the manuscript technically sound, and do the data support the conclusions?

Reviewer #1: Yes

2. Has the statistical analysis been performed appropriately and rigorously? 

Reviewer #1: Yes

3. Have the authors made all data underlying the findings in their manuscript fully available?

Reviewer #1: Yes

4. Is the manuscript presented in an intelligible fashion and written in standard English?

Reviewer #1: Yes

5. Review Comments to the Author

Reviewer #1: Anjos et al. investigated the genetic diversity and population structure using high throughput genotyping in teak; and confirmed the existence of vast variability. The paper consists of sufficient originality and scientific quality; therefore, it is suitable to publish in the journal after minor corrections as follows:

1. While the overall language and clarity are reasonable, there are some instances where the wording could be improved for better readability. The author must check the manuscript's grammatical and typological mistakes throughout the manuscripts.

Line 25: “the sequencings ware aligned” or “the sequencings were aligned”.

Line 480-481: clear the sentence “Based in our analyses true high thought genotyping by sequence with a representative pool of teak genotypes…”

2. Check the figure’s serial numbers.

Line 202 and 212: The figure number is same for both the pictures, i.e. Fig 3.

3. The discussion thoroughly analyses and compares the findings with previous studies. However, it would be beneficial to include a more extensive discussion on the implications of the narrow genetic base of Tectona grandis. What are the potential consequences for the species' adaptability, resilience, and future breeding programs?

4. The conclusion could be strengthened by succinctly summarising the essential findings and their implications. It would be helpful to restate the main objectives and emphasise the novelty and significance of the research.

5. Follow the proper journal format for writing the citations in the text and the reference section of the manuscript. The author can use the PLOS template to prepare references. Some mistakes are

Line 109: “test (Keiding et al., 1986; Kjaer & Suangtho, 1995)), while those of some unregistered”

Line 133: “(https://zenodo.org/record/3962665#.Y7MxfezMKrM)”

6. PLOS authors have the option to publish the peer review history of their article (what does this mean?). If published, this will include your full peer review and any attached files.

Reviewer #1: **Yes: **Sarfraz Ahmad

---

## [Author Response · Author response to Decision Letter 0]

1 Sep 2023

FROM: Thiago Alexandre Santana Gilio , Corresponding Author

TO: Dr. Mehdi Rahimi

 PlosOne Academic Editor

Dear Dr. Mehdi Rahimi

I am delighted to resubmit our original research article titled "Reassessing the Genetic Variability of Tectona grandis through High-Throughput Genotyping: Insights on its Narrow Genetic Base" for your consideration for publication in the Plos One journal, incorporating the necessary revisions as requested.

I emphasize again that our manuscript is the first to use sequencing genotyping in Tectona grandis, and through our study we concluded that the genetic diversity in this species is greater than that reported in other previously published manuscripts.

Outlined below are responses addressing the feedback provided by the Academic Editor and the Reviewer:

Review responses to the Academic Editor

Response - Manuscript has been amended to meet journal style and file naming guidelines as requested.

2. Response - We have modified the acknowledgments section as requested to exclude the funding information. 

Please amend the Financing Statement to the new version:

“This study was partially funded by the Coordination for the Improvement of Higher Education Personnel – Brazil (CAPES), by the Mato Grosso State Research Support Foundation (FAPEMAT) and by Teak Resources Company (TRC agroforestry) LTDA”

Response - The financial disclosure will be revised before submission..

4-5. The data availability statement was changed and the DOI of the publication was inserted at the end of the manuscript (DOI:10.5281/zenodo.8171369)

6. We note that you have referenced (Keogh R. Teak (Tectona grandis) provenances of the Caribbean, Central America, Venezuela and Colombia. Unpublished Original document at FAO headquarters, Rome. 1978.) which has currently not yet been accepted for publication. Please remove this from your References and amend this to state in the body of your manuscript: (ie “Bewick et al. [Unpublished]”) as detailed online in our guide for authors

Response - The highlighted reference was removed from the manuscript, without prejudice to the text.

7. Issues about the Figures.

Response - The highlighted figure was removed from the manuscript due to the copyright situation, without prejudice to the text.

Response: Supplemental file legends have been included in the supporting information at the end of the manuscript, as requested.

Response: the references have been refreshed, along with the update of all citation numbers throughout the manuscript.

The citation “(Keogh R. Teak (Tectona grandis) from the Caribbean, Central America, Venezuela and Colombia. Original document unpublished at FAO headquarters, Rome. 1978.)” was removed from the text without bias, taking into account the incorporation of other jointly used sources.

The subsequent references have been included in the Discussion section and the reference list to substantiate a point requested by the reviewer: 

63. Reis CAF, Paludzyszyn Filho E. Estado da arte de plantios com espécies florestais de interesse para o Mato Grosso. Documentos - Embrapa Florestas. 2011 [cited 16 Aug 2023]. Available: https://www.infoteca.cnptia.embrapa.br/bitstream/doc/898075/1/Doc215.pdf

64. RAMALHO MAP, Santos JB dos, Pinto C, Souza EA de, Gonçalves FMA, Souza JC de. Genética na agropecuária. rev. Lavras: Ufla. 2004. 

65. Carvalho J, SILVA MM de A, Medeiros MJL. Perda e conservação dos recursos genéticos vegetais. Documentos - Embrapa Algodão. 2009 [cited 16 Aug 2023]. Available: https://www.infoteca.cnptia.embrapa.br/bitstream/doc/656849/1/DOC221.PDF

Responses to the Reviewer

1. While the overall language and clarity are reasonable, there are some instances where the wording could be improved for better readability. The author must check the manuscript's grammatical and typological mistakes throughout the manuscripts.

Line 25: “the sequencings ware aligned” or “the sequencings were aligned”.

Line 480-481: clear the sentence “Based in our analyses true high thought genotyping by sequence with a representative pool of teak genotypes…”

Response: Thank you very much for your input. The highlighted grammatical errors have now been rectified.

2. Check the figure’s serial numbers.

Line 202 and 212: The figure number is same for both the pictures, i.e. Fig 3.

Response: The numbering of figures has been adjusted to align with the recent modifications.

3. The discussion thoroughly analyses and compares the findings with previous studies. However, it would be beneficial to include a more extensive discussion on the implications of the narrow genetic base of Tectona grandis. What are the potential consequences for the species' adaptability, resilience, and future breeding programs?

Response: Two paragraphs have been inserted in the Discussion on this point section.

4. The conclusion could be strengthened by succinctly summarising the essential findings and their implications. It would be helpful to restate the main objectives and emphasise the novelty and significance of the research.

Response: Conclusion section has been amended as requested to summarize the findings.

5. Follow the proper journal format for writing the citations in the text and the reference section of the manuscript. The author can use the PLOS template to prepare references. Some mistakes are

Line 109: “test (Keiding et al., 1986; Kjaer & Suangtho, 1995)), while those of some unregistered”

Line 133: “(https://zenodo.org/record/3962665#.Y7MxfezMKrM)”

Response: The manuscript citations have been changed to journal format.

The authors would like to seize this opportunity to confirm that all figures have been submitted for PACE digital diagnosis, as recommended.

We are confident that our manuscript is well-suited for publication in Plos One and will contribute significantly as a point of reference within the field. Additionally, it has not been previously published, nor is it presently under review for publication elsewhere. There are no conflicts of interest to declare.

We extend our gratitude for your thoughtful consideration of our submission.

Prof. Dr. Thiago Alexandre Santana Gilio and Dra. Isabela Vera dos Anjos

---

## [Decision Letter · Decision Letter 1]

20 Sep 2023

Reassessing the Genetic Variability of Tectona grandis through high-throughput genotyping: Insights on its Narrow Genetic Base

PONE-D-23-12305R1

Dear Dr. Gilio,

We’re pleased to inform you that your manuscript has been judged scientifically suitable for publication and will be formally accepted for publication once it meets all outstanding technical requirements.

Kind regards,

Mehdi Rahimi, Ph.D.

Academic Editor

PLOS ONE

Additional Editor Comments (optional):

Reviewers' comments:

Reviewer's Responses to Questions

**Comments to the Author**

1. If the authors have adequately addressed your comments raised in a previous round of review and you feel that this manuscript is now acceptable for publication, you may indicate that here to bypass the “Comments to the Author” section, enter your conflict of interest statement in the “Confidential to Editor” section, and submit your "Accept" recommendation.

Reviewer #1: All comments have been addressed

2. Is the manuscript technically sound, and do the data support the conclusions?

Reviewer #1: Yes

3. Has the statistical analysis been performed appropriately and rigorously? 

Reviewer #1: Yes

4. Have the authors made all data underlying the findings in their manuscript fully available?

Reviewer #1: Yes

5. Is the manuscript presented in an intelligible fashion and written in standard English?

Reviewer #1: Yes

6. Review Comments to the Author

Reviewer #1: (No Response)

7. PLOS authors have the option to publish the peer review history of their article (what does this mean?). If published, this will include your full peer review and any attached files.

Reviewer #1: **Yes: **Sarfraz Ahmad

---

## [Editor Report · Acceptance letter]

16 Oct 2023

PONE-D-23-12305R1 

Reassessing the Genetic Variability of *Tectona grandis* through high-throughput genotyping: Insights on its Narrow Genetic Base 

Dear Dr. Gilio:

I'm pleased to inform you that your manuscript has been deemed suitable for publication in PLOS ONE. Congratulations! Your manuscript is now with our production department. 

Kind regards, 

on behalf of

Associate Prof. Mehdi Rahimi 

Academic Editor

PLOS ONE